# Prediction of Stroke Outcome Using Natural Language Processing-Based Machine Learning of Radiology Report of Brain MRI

**DOI:** 10.3390/jpm10040286

**Published:** 2020-12-16

**Authors:** Tak Sung Heo, Yu Seop Kim, Jeong Myeong Choi, Yeong Seok Jeong, Soo Young Seo, Jun Ho Lee, Jin Pyeong Jeon, Chulho Kim

**Affiliations:** 1Department of Convergence Software, Hallym University, Chuncheon 24252, Korea; gjxkrtjd221@gmail.com (T.S.H.); yskim.hallym@gmail.com (Y.S.K.); jeong5905@gmail.com (J.M.C.); dnfkdi1995@gmail.com (Y.S.J.); syseo96@gmail.com (S.Y.S.); 2Department of Otorhinolaryngology and Head and Neck Surgery, Chuncheon Sacred Heart Hospital, Chuncheon 24253, Korea; zoonox@nate.com; 3Department of Neurosurgery, Chuncheon Sacred Heart Hospital, Chuncheon 24253, Korea; jjs6553@hanmail.net; 4Department of Neurology, Chuncheon Sacred Heart Hospital, Chuncheon 24253, Korea

**Keywords:** ischemic stroke, functional outcome, machine learning, deep learning, natural language processing, magnetic resonance imaging

## Abstract

Brain magnetic resonance imaging (MRI) is useful for predicting the outcome of patients with acute ischemic stroke (AIS). Although deep learning (DL) using brain MRI with certain image biomarkers has shown satisfactory results in predicting poor outcomes, no study has assessed the usefulness of natural language processing (NLP)-based machine learning (ML) algorithms using brain MRI free-text reports of AIS patients. Therefore, we aimed to assess whether NLP-based ML algorithms using brain MRI text reports could predict poor outcomes in AIS patients. This study included only English text reports of brain MRIs examined during admission of AIS patients. Poor outcome was defined as a modified Rankin Scale score of 3–6, and the data were captured by trained nurses and physicians. We only included MRI text report of the first MRI scan during the admission. The text dataset was randomly divided into a training and test dataset with a 7:3 ratio. Text was vectorized to word, sentence, and document levels. In the word level approach, which did not consider the sequence of words, and the “bag-of-words” model was used to reflect the number of repetitions of text token. The “sent2vec” method was used in the sensation-level approach considering the sequence of words, and the word embedding was used in the document level approach. In addition to conventional ML algorithms, DL algorithms such as the convolutional neural network (CNN), long short-term memory, and multilayer perceptron were used to predict poor outcomes using 5-fold cross-validation and grid search techniques. The performance of each ML classifier was compared with the area under the receiver operating characteristic (AUROC) curve. Among 1840 subjects with AIS, 645 patients (35.1%) had a poor outcome 3 months after the stroke onset. Random forest was the best classifier (0.782 of AUROC) using a word-level approach. Overall, the document-level approach exhibited better performance than did the word- or sentence-level approaches. Among all the ML classifiers, the multi-CNN algorithm demonstrated the best classification performance (0.805), followed by the CNN (0.799) algorithm. When predicting future clinical outcomes using NLP-based ML of radiology free-text reports of brain MRI, DL algorithms showed superior performance over the other ML algorithms. In particular, the prediction of poor outcomes in document-level NLP DL was improved more by multi-CNN and CNN than by recurrent neural network-based algorithms. NLP-based DL algorithms can be used as an important digital marker for unstructured electronic health record data DL prediction.

## 1. Introduction

Stroke is one of the leading causes of disability in developing and developed countries. According to the 2016 Global Burden of Disease, Injuries and Risk Factors Study report, strokes caused 5.5 million deaths and 116 million disability-adjusted life-years annually [1]. A stroke is not a communicable disease and can be prevented through education and the management of risk factors, such as hypertension, diabetes, dyslipidemia, and atrial fibrillation. In addition, intensive treatment of patients whose stroke prognosis is expected to be poor can improve the prognosis. Therefore, predicting the prognosis of a stroke is important in rapid treatment decisions and the effective distribution of medical resources.

Recently, machine learning (ML) or deep learning (DL) strategies have been used to predict stroke outcomes more accurately than did conventional logistic models. Lin et al. reported an ML method using 206 clinical variables that could reach an area under the receiver operating characteristics (AUROC) of 0.94 in predicting the 90-day functional outcome of ischemic and hemorrhagic stroke patients [2]. Heo et al. reported the efficacy of DL algorithms over the Acute Stroke Registry and Analysis of Lausanne (ASTRAL) score—which is a widely used logistic regression-based algorithm for stroke outcome prediction [3]—to predict the poor functional outcome in patients with acute ischemic stroke (AIS). DL algorithms showed considerably better performance than traditional prediction models did in predicting the prognosis of stroke patients using numerical data. In addition, DL algorithms using brain magnetic resonance imaging (MRI) showed improved accuracy in predicting the final infarct volume and reperfusion status [4]. Hilbert reported that computed tomography angiography DL using ResNet and an autoencoder could provide well-performing image biomarkers for predicting the functional outcomes in AIS patients who received endovascular treatment [5].

Electronic health records (EHRs) are mainly composed of unstructured data. Among them, unstructured text data such as a doctor’s note, nursing records, and radiology and pathology text reports, account for the largest proportion of the EHR [6]. Natural language processing (NLP) is a well-known artificial intelligence technology, from which valuable information for proper diagnosis, treatment strategy, and outcome prediction can be obtained using text vectorization and DL algorithms. In the field of stroke research, NLP and ML algorithms have proven their efficacy in differentiating the specific diagnosis and image phenotype of a stroke [7,8,9,10,11]. However, only a few studies have been conducted by utilizing radiology text reports of brain MRI to predict future functional outcomes or certain phenotypes using NLP and ML algorithms. Therefore, we aimed to assess whether brain MRI text analysis using NLP and ML algorithms can predict a 3-month functional outcome in AIS patients.

## 2. Materials and Methods

### 2.1. Study Participants

We used a prospectively collected stroke database from a tertiary academic hospital, in which all patients’ records—including demographic, clinical, laboratory, and radiology reports—were regularly collected and audited by trained stroke practitioners. From January 2014 to December 2019, a total of 2538 AIS patients were eligible for this study. We excluded patients with a history of strokes (n = 563) because we analyzed the impact of brain MRI text reports, which usually included text information for acute structural lesions and previous structural abnormalities. In other words, we excluded patients with previous strokes because the importance of text data regarding recent acute lesions may be reduced with an increase in the number of text descriptions of previous lesions. In addition, patients without proper magnetic resonance (MR) images (n = 135) were also excluded. This study was approved by the Chuncheon Sacred Heart Hospital Institutional Review Board/Ethics Committee (IRB No. 2019-I064). Written informed consent for the registry enrollment and 3-month outcome capture was provided by the participants or their guardians.

### 2.2. Data Collection Using MRI Radiology Reports

When patients with acute cerebral infarction are hospitalized, performing multiple MRI scans is a common clinical practice to define the dynamic status of ischemia and vessel occlusion. Therefore, we limited the data to be used for text analysis to the MRI text report of the first MRI scan during admission. Conventional stroke MRI sequences included T1/T2-weighted sequence, apparent diffusion coefficient and diffusion-weighted sequence, fluid-attenuated inversion recovery sequence, gradient echo sequence, time-of-flight, and gadolinium-enhanced angiography sequences [12]. We used 3.0T brain MRI scanner (Skyra, Siemens, Germany) with the following parameters: TR, 9000–11,000 range; TE 120–130 range; matrix size, 256 × 256; field of view, 230 × 230 mm; slice thickness, 5mm; inter-slice gap, 1mm. During the study period, one neuroradiologist interpreted the brain MRI and reported the narrative descriptions and final conclusions in the text data. Examples of brain MRI radiology reports are shown in Appendix A. We only used narrative descriptions of the MRI images and removed clinical information and conclusions.

### 2.3. NLP and ML Algorithms

As described earlier, we only used the description texts of the brain MRI radiology reports to predict the 3-month poor outcomes. All text data were parsed into vectors of different levels. We used the quanteda R package and the NLTK Python package to tokenize the MRI text data.

#### 2.3.1. Word-level Approach

First, all texts were parsed into word vectors, in which one word became a one word “token” vector. All texts were changed into tokens using lowercase alphabets, and stop words, punctuation, symbols, and hyphens were removed. Each word token was entered into the ML algorithms as a “bag-of-words” (BOW) model (Figure 1) [13]. In addition, we applied DL algorithms to classify MRI texts for good and poor outcomes. The detailed architecture of the DL algorithms for the word-level approach is shown in Appendix A.

#### 2.3.2. Sentence-Level Approach

In a sentence-level approach, a patient’s brain MRI document is divided into sentence units and entered into a vector. Unlike the word embedding method, the meaning of a sentence is expressed as a vector using the sent2vec sentence embedding library. Then, the vectored sentences perform classification tasks by concatenating dense layers.

#### 2.3.3. Document-Level Approach

In this method, the entire MRI reading text corresponding to one patient was used as an input to the ML algorithms. All the words in a document were vectorized using BioWordVec [14], which is a pretrained biomedical FastText embedding model that includes the meaning of words in 200 dimensions.

### 2.4. Primary Outcome Measure

All the patients visited outpatient clinics after hospital discharge to evaluate stroke outcomes. The functional outcomes were evaluated using the modified Rankin Scale (mRS) score 3 months after the onset of stroke symptoms—with scores ranging from 0 for “no stroke-related symptoms” to 6 for “stroke-related death” [15]—and mRS scores of 3–6 indicated a poor stroke prognosis. In almost all stroke clinical trials, the assessment of functional recovery of the stroke patient for drugs or interventions is evaluated with mRS at 3 months after the stroke symptom onset. Therefore, we used 3-month mRS as a main outcome to predict using the brain MRI text. Our goal was to identify which ML algorithm was superior in predicting 3-month poor outcomes using these admission brain MRI texts.

### 2.5. ML Task

All MRI texts were randomly divided into training and test datasets in a ratio of 7:3, in which the proportion of poor outcome MRI texts was similar. After dividing the training data into 5-folds, model training was performed in 4-folds, the model was validated using the remaining 1-fold training data, and the model’s performance was measured using the other test dataset. In the word-level approach, we used the least absolute shrinkage and selection operator (LASSO) regression, single decision tree, random forest (RF), and support vector machine (SVM) techniques for the ML algorithm. For this ML task, we extracted the feature importance of the text vectors in the RF classifier to identify which tokens were important for predicting poor outcomes in MRI texts [16]. In addition, convolutional neural network (CNN), multilayer perceptron (MLP), long short-term memory (LSTM), and bidirectional LSTM (bi-LSTM) CNN&LSTM techniques were employed as DL algorithms to predict poor outcomes in MRI texts. In the document-level approach, all document vectors preprocessed with word embedding were then applied to the CNN, multi-kernel CNN (multi-CNN), LSTM, and bi-LSTM algorithms to predict the stroke prognosis. The detailed architecture of the DL algorithms is depicted in Appendix A. The grid search technique was used to optimize the best hyperparameters of each algorithm [17]. Model training was performed on TensorFlow and Keras platform using NVIDIA’s GeForce GTX 1080ti graphics processing units, dual Xeon central processing units, and 128 GB RAM server. Table 1 shows the summarized ML algorithms used in different level of text vectorization approach.

### 2.6. Statistical Methods

Baseline characteristics of the patients—for poor and good outcomes—were compared using the Student’s *t*-test, Mann–Whitney U, or Pearson’s χ^2^-test, as appropriate. To identify which words were observed more in the MRI report of patients with poor outcomes than in those with good outcomes, we performed the χ^2^-test, and the results were provided as a keyness plot [18]. The performance of each ML classifier was evaluated using unseen test data. We calculated the probability score of the data in each ML algorithm, and the performance of each ML algorithm was measured using the AUROC curve. ML classification tasks were performed with R version 3.6.1 (the R Foundation for Statistical Computing) and Python version 3.7.7 in the anaconda environment.

## 3. Results

A total of 1840 MRI text reports were included in the final analysis. The proportion of poor outcomes in the training and test datasets were 36.7% and 33.0%, respectively. Comparisons of the clinical characteristics of the participants between poor and good outcomes are presented in Table 2. There were no differences in the clinical and demographic variables between the training and test datasets.

Appendix A shows the most frequently observed tokens in the training and test datasets—word tokens from the MRI texts being similarly distributed in both. The keyness plot (Figure 2A) shows that several tokens that described large territory infarct lesions were frequently observed in the poor outcome brain MRI texts. Using the RF algorithm, we identified which tokens were important for predicting poor outcome MRI texts (Figure 2B). In this variable importance plot, several tokens that described large territory involvement in brain MRI texts (middle cerebral artery (mca); intravascular; territori) were the most important tokens for predicting poor outcome brain MRI texts.

### Performance of ML Algorithm for Poor Outome Brain MRI Texts

Appendix A shows the comparison of input tokens between the training and test datasets using the BOW model. There were no significant input features between them. Figure 3A shows the performance of the LASSO regression, single decision tree, SVM, and RF algorithms in predicting poor outcome brain MRI texts. The RF algorithm had the best AUROC (0.782) score among them. Figure 3B shows the result of the BOW DL model, in which the CNN’s performance (0.769) was better than that of the other DL algorithms. However, the RF algorithm was the best classifier in predicting poor outcome brain MRI texts using a word-level approach.

Figure 3C shows the results of the sentence- and document-level DL models. The sentence-level approach was not superior to the document-level approach in predicting poor outcomes. Overall, the document-level approach exhibited better performance than did the word- or sentence-level approaches. Among all the ML classifiers, the multi-CNN’s performance was the best (0.805), followed by the CNN (0.799) algorithm.

## 4. Discussion

In this study on the prediction of poor outcomes using NLP-based DL of radiology text reports, we identified that the multi-CNN DL algorithm can successfully predict future outcomes in patients with AIS. In particular, document-level vectorization of the entire radiology text reports showed better performance than did the word- or sentence-level NLP approaches. Although the DL algorithms did not show which word vectors were important for this classification, other ML procedures such as the RF could identify important text features from the brain MRI texts to predict future outcomes using ML algorithms.

MRI is essential for sophisticated stroke diagnosis, and we can predict the future outcome of AIS patients to some extent through certain imaging findings such as distal hyperintensity vessel sign [19], diffusion restrictive infarct volume [20], or hemorrhagic transformation lesion [21] in MR images. However, MRI radiology text reports are often described more freely than with a structured format and are difficult to quantify because a radiologist’s subjective judgment is reflected in the text rather than the MR image itself. However, we found that NLP-based DL of free text MRI reports taken only once (at admission) could successfully predict the prognosis of AIS patients.

We found that DL using a document-level approach was a better method for predicting the prognosis of AIS patients using brain MRI free text reports than using word- or sentence-level approaches. Several NLP-based ML studies have shown good classification performance in differentiating certain disease phenotypes from the corresponding free text MRI reports with a word-level approach alone [22,23,24,25]. However, the studies did not implement DL algorithms to predict their own targets. NLP processing was also found to have performed analyses using only the BOW model. Our findings showed better performance in the document-level approach—which understands the sequence of sentences as a whole—than the BOW model—which interprets each word as a machine-readable vector. However, the performance of the CNN and multi-CNN algorithms using a document-level approach was superior to that of the recurrent neural network (RNN) approach—which better reflects the sequence of texts or signals, using LSTM and bi-LSTM methods [26,27]. This suggests the possibility that a CNN’s performance is better than that of an RNN because the structural vectorization of the sequential sentences is already reflected in the document-level approach. The advantages of CNNs over RNNs—such as LSTM and gated recurrent units—are that they have a smaller number of parameters, thus delivering good computational speed and affording a more efficient setup of convolutional layers to learn local information than is the case with RNNs [28,29]. In addition, the CNN needs to construct an algorithm into a deeply layered architecture for the machine to learn the whole sequence of sentences efficiently, and this increase in layers can lead to vanishing gradients or exploding problems in the CNN algorithm. In this study, however, we suggested that the classification performance of the CNN algorithm might have been better than that of the RNN because the CNN architecture used in the document-level approach did not have many layers, which in turn did not require learning for the local minima. In other words, we have shown that when predicting a certain phenotype using ML of brain MRI texts using a document-level approach, it is sufficient to use CNNs rather than RNNs, which require significant computing power.

There are numerous risk predictors for poor outcomes in AIS patients. First, conventional risk factors and relevant laboratory results, such as hypertension, diabetes, dyslipidemia, hyperglycemia, and blood pressure, are important biomarkers for poor outcome predictions [30]. As multimodal MRI for AIS has been widely used, collateral vessel signs [31], diffusion-weighted lesion volumes [32], cerebral microbleeds [33], and combined hemorrhagic transformation lesions [34], are well-known as important image markers for poor outcomes. In recent years, DL algorithms for MRI have also been a useful for treatment effect of AIS patients [35]. However, there have been no reports of text biomarkers for the EHRs of stroke patients. We showed that this text vector of image reports could also be sufficiently useful for outcome prediction.

DL generally shows superior performance over conventional prediction models. This study also showed that the performances of DL using CNN and RNN was better than that of ML methods such as RF and SVM [36]. However, we did not determine what type of text vectors could predict the poor prognosis of AIS patients in NLP-based text DL tasks. However, ML algorithms such as RF can show which factors are important in classification/regression prediction tasks, as shown in Appendix A. Similar to the infarct volume in the stroke MR image, the variable importance plot of the poor outcome brain MRI texts showed that a combination of stemming words depicting large territory lesions such as “mca, territori, intravascular (thrombosis), and complet (occlusion)” are important features in differentiating poor outcome brain MRI text reports from those with a good outcome. Although the DL algorithm has evolved more than conventional ML algorithms, ML methods such as RF or SVM techniques in text-based prediction are still important in identifying important “digital phenotypes” in vast unconstructed EHR text data and converting them into structured data.

Our study has some limitations. First, because the MRI text report was read by neuroradiologists in one hospital, an external validation using data from another hospital is needed to determine whether the DL algorithm would demonstrate the same performance in predicting stroke outcomes. Second, we performed NLP-based DL only on English texts. The sequential combination of words is considerably important for understanding sentences in the case of inflectional languages such as English. However, the type of prefix added to the word root is more important than the order of words in the case of agglutinative languages such as Korean, Turkish, and Japanese. Therefore, we cannot conclude whether these DL algorithms will perform better in predicting poor outcomes using brain MRI text reports in languages other than English. Despite these limitations, our study has some strengths. First, we used prospectively collected stroke outcomes from stroke physicians and certified nurses. This prospective clinical outcome capture strategy guaranteed the performance of our ML model. We did not classify the findings of brain MRI from the text data but predicted the future clinical outcome of the patient. We believe that this study may have provided some insights for conducting further studies using EHR information that could predict future clinical events.

## 5. Conclusions

In this study, we found that when predicting future clinical outcomes using NLP-based ML of brain MRI radiology free text reports, DL algorithms showed superior performance over other ML algorithms. In particular, the prediction of poor outcomes in document-level NLP DL was improved more by using multi-CNN and CNN than by using RNN-based algorithms. In later, unstructured EHR text, data can be widely used for extracting and predicting important clinical outcomes using an NLP-based DL strategy.

## Figures and Tables

**Figure 1 jpm-10-00286-f001:**
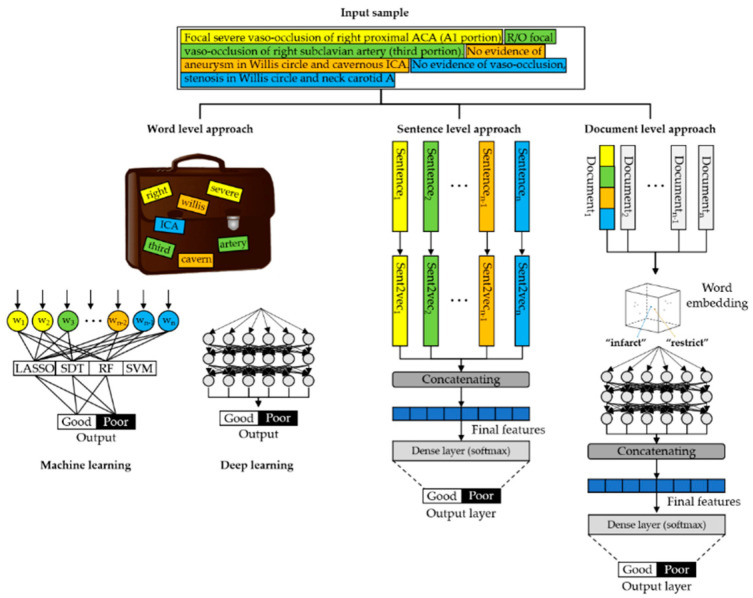
Example of word-, sentence-, and document-level approaches using natural language preprocessing and machine learning of brain MRI radiology text reports (LASSO, least absolute shrinkage and selection operator; SDT, single decision tree; RF, random forest; SVM, support vector machine).

**Figure 2 jpm-10-00286-f002:**
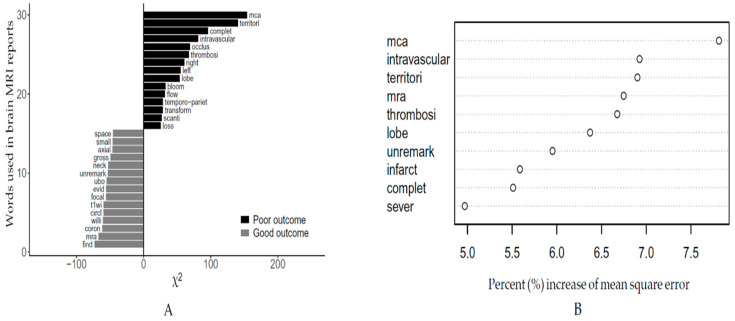
Result of keyness plot showing the relative frequency of word tokens between the good and poor outcome groups (**A**). Variable importance plot for the stemming words in the random forest algorithm (mca, middle cerebral artery) (**B**).

**Figure 3 jpm-10-00286-f003:**
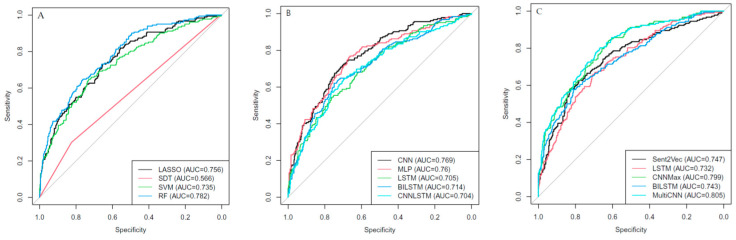
Results of the area under the receiver operating characteristic curves showing the performance of the word-level ML approach (**A**), DL approach (**B**), and (**C**) sentence- or document-level DL approach to predict poor outcome brain MRI texts (LASSO, least absolute shrinkage and selection operator; SDT, single decision tree; SVM, support vector machine; RF, random forest; CNN, convolutional neural network; MLP, multilayer perceptron; LSTM, long short-term memory; BILSTM, bidirectional LSTM; Sent2vec, CNN-Max, max-pooling CNN; Multi-CNN, multi-kernel CNN; AUC, area under the receiver operating characteristic curves).

**Table 1 jpm-10-00286-t001:** Type of machine learning algorithm used in different level of natural language processing approach. The filled cell indicates the machine learning algorithm used. (ML, machine learning; LASSO, least absolute shrinkage and selection operator; SDT, single decision tree; RF, random forest; SVM, support vector machine; CNN, convolutional neural network; MLP, multilayer perceptron; LSTM, long short-term memory; Bi-LSTM, bidirectional LSTM; CNN-Max, max-pooling CNN; Multi-CNN, multi-kernel CNN).

Type of ML Algorithm		Level of Approach	
	Word Level	Sentence Level	Document Level
LASSO regression			
SDT			
RF			
SVM			
CNN			
MLP			
LSTM			
Bi-LSTM			
CNN&LSTM			
CNNmax			
Multi-CNN			

**Table 2 jpm-10-00286-t002:** Baseline characteristics of the total study population.

	Training (*n* = 1288)	Test (*n* = 522)	*p* Value
Age, years	69.3 ± 12.7	69.1 ± 12.8	0.773
Male, %	736 (57.1)	321 (58.2)	0.726
Height, cm	165.0 ± 13.1	164.4 ± 67.5	0.603
Weight, kg	68.5 ± 12.7	67.3 ± 13.5	0.698
NIHSS scale, mg/dL	4.8 ± 5.6	4.4 ± 5.3	0.226
Risk factors			
Hypertension	835 (64.8)	353 (63.9)	0.758
Diabetes	420 (32.6)	198 (35.9)	0.192
Dyslipidemia	228 (17.7)	94 (17.0)	0.779
Current smoking	301 (23.4)	130 (23.6)	0.981

Values are presented as mean ± standard deviation or number (column percent) as appropriate. NIHSS, National Institute of Health Stroke Scale.

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
