# Peer review of "Prediction of Stroke Outcome Using Natural Language Processing-Based Machine Learning of Radiology Report of Brain MRI"

_jpm, 2020, doi:10.3390/jpm10040286_

Round 1
Reviewer 1 Report
This study employed several machine learning algorithms using brain MRI free-text reports of acute ischemic stroke patients to predict poor outcomes, and accessed the performance of each algorithm. The manuscript is well written. The analysis the author did is straightforward. I have a few comments:
- could the author please provide a summary table in method, indicating what machine learning algorithms were used in word, sentence, and document levels, respectively?
- the authors used the MRI reports to predict the 3-month poor outcomes? Why chose “3-month”, not “6-month” or other? If the author compared the prediction performance of different model in “3-month”, and, for example, “6-month”?
Author Response
Thank you for your review. We have tried our best to reflect your suggestion as below.
This study employed several machine learning algorithms using brain MRI free-text reports of acute ischemic stroke patients to predict poor outcomes, and accessed the performance of each algorithm. The manuscript is well written. The analysis the author did is straightforward. I have a few comments:
- could the author please provide a summary table in method, indicating what machine learning algorithms were used in word, sentence, and document levels, respectively?
; Thank you for the comment. As the reviewer’s suggestion, we added summarized table in Method section.
- the authors used the MRI reports to predict the 3-month poor outcomes? Why chose “3-month”, not “6-month” or other? If the author compared the prediction performance of different model in “3-month”, and, for example, “6-month”?
; Thanks for the comment. In all stroke clinical trials, the assessment of functional recovery of the stroke patient for drugs or interventions is evaluated with mRS at 3 months after the stroke symptom onset. Therefore, we used 3-month mRS as an important clinical outcome in our study. To help our readers understanding, we have added the following sentences.
In almost stroke clinical trials, the assessment of functional recovery of the stroke patient for drugs or interventions is evaluated with mRS at 3 months after the stroke symptom onset. Therefore, we used 3-month mRS as a main outcome to predict using the brain MRI text.

Reviewer 2 Report
The paper written by the following Authors: Tak Sung Heo, Jeong Myeong Choi, Yeong Seok Jeong, Soo Young Seo, Yu Seop Kim, Jun Ho Lee, Jin Pyeong Jeon and Chulho Kim, entitled “Prediction of Stroke Outcome Using Natural 2 Language Processing-based Machine Learning of 3 Radiology Report of Brain MRI” presents an interesting study on a novel approaches to brain MRI text analysis using NLP and ML algorithms.
Although the paper is interesting, I have some major concerns:
Title
The title reflects the results presented here.
Abstract
The abstract is lacking the aim of the material and methods description as well as an informative conclusion. It should be written in more details.
Introduction
In the introduction part Authors should add some overall information in paragraph/paragraphs dedicated on numerical and experimental methods applied in medical diagnosis/treatment. The authors should consider justifying the motivation of this study with recent study and e.g. cite the paper listed below:
Polanczyk et al. Artificial Circulatory Model for Analysis of Human and Artificial Vessels, https://doi.org/10.3390/app8071017
Material and Methods
There is no information describing analyzed patients. It should be included in the manuscript.
There is no information about the resolution of MRI data. It should be included in the manuscript.
There is no information about the hardware applied in the analysis. It should be included in the manuscript.
Conclusions
More informative conclusions should be included.
Figures
Figure 2 – the resolution should be improved.
Figure 3 – the resolution should be improved.
Author Response
Reviewer #2.
Thank you for your review. We have tried our best to reflect your suggestion as below.
The paper written by the following Authors: Tak Sung Heo, Jeong Myeong Choi, Yeong Seok Jeong, Soo Young Seo, Yu Seop Kim, Jun Ho Lee, Jin Pyeong Jeon and Chulho Kim, entitled “Prediction of Stroke Outcome Using Natural Language Processing-based Machine Learning of Radiology Report of Brain MRI” presents an interesting study on a novel approaches to brain MRI text analysis using NLP and ML algorithms.
Although the paper is interesting, I have some major concerns:
Title
The title reflects the results presented here.
; Thanks for the comment. We used conventional ML algorithms (RF, SVM, LASSO regression, single decision tree) and (CNN, Multilayer perceptron, and LSTM) to predict poor outcomes of brain MRI text reports. We tried to write the title informative and concisely. If you have any recommendations regarding the change of a better title, we will reflect them immediately.
Abstract
The abstract is lacking the aim of the material and methods description as well as an informative conclusion. It should be written in more details.
; Thanks for the comment. As the reviewer’s suggestion, we added more materials in the Abstract section as below:
Brain magnetic resonance imaging (MRI) is useful for predicting the outcome of patients with acute ischemic stroke (AIS). Although deep learning (DL) using brain MRI with certain image biomarkers has shown satisfactory results in predicting poor outcomes, no study has assessed the usefulness of natural language processing (NLP)-based machine learning (ML) algorithms using brain MRI free-text reports of AIS patients. Therefore, we aimed to assess whether NLP-based ML algorithms using brain MRI text reports could predict poor outcomes in AIS patients. This study included only English text reports of brain MRI examined during admission of AIS patients. Poor outcome was defined as a modified Rankin Scale score of 3–6, and the data were captured by trained nurses and physicians. We only included MRI text report of the first MRI scan during the admission. The text dataset was randomly divided into a training and test dataset with a 7:3 ratio. Text was vectorized to word, sentence, and document levels. In word level approach, which did not consider the sequence of words, and the "bag-of-words" model was used to reflect the number of repetitions of text token. The "sent2vec" method was used in the sensation-level approach considering the sequence of words, and the word embedding was used in document level approach. In addition to conventional ML algorithms, DL algorithms such as the convolutional neural network (CNN), long short-term memory, and multilayer perceptron were used to predict poor outcomes using 5-fold cross-validation and grid search techniques. The performance of each ML classifier was compared with the area under the receiver operating characteristic (AUROC) curve. Among 1840 subjects with AIS, 645 patients (35.1%) had a poor outcome 3 months after the stroke onset. Random forest was the best classifier (0.782 of AUROC) using a word-level approach. Overall, the document-level approach exhibited better performance than did the word- or sentence-level approaches. Among all the ML classifiers, the multi-CNN algorithm demonstrated the best classification performance (0.805), followed by the CNN (0.799) algorithm. When predicting future clinical outcomes using NLP-based ML of radiology free-text reports of brain MRI, DL algorithms showed superior performance over the other ML algorithms. In particular, the prediction of poor outcomes in document-level NLP DL was improved more by multi-CNN and CNN than by recurrent neural network-based algorithms. In later, NLP-based DL algorithms can be used important digital marker for unstructured electronic health record data DL.
Introduction
In the introduction part Authors should add some overall information in paragraph/paragraphs dedicated on numerical and experimental methods applied in medical diagnosis/treatment. The authors should consider justifying the motivation of this study with recent study and e.g. cite the paper listed below:
Polanczyk et al. Artificial Circulatory Model for Analysis of Human and Artificial Vessels, https://doi.org/10.3390/app8071017
; Thanks for the comment. We tried to investigate whether text data DL can predict the clinical outcome of stroke patients in this paper. In “Introduction” section, we tried to make the background as concise as possible about whether clinical prediction of stroke is important and whether deep learning using EHR text data can predict clinical outcomes well. However, in those paper recommended by the reviewer to quote, the content of the paper was not related with stroke, deep learning, and natural language processing (keyword of our paper). Therefore, we did not quote those paper for the brevity of the introduction.
Material and Methods
There is no information describing analyzed patients. It should be included in the manuscript.
; Thanks for the comment. We did not use clinical information of patients, but use MRI text only to predict patient’ clinical outcomes. At first, We did not include clinical information related to AIS patients because it seems to be inappropriate to include clinical information of the patients. However, we finally presented clinical information in Table 2, as the reviewer pointed out.
There is no information about the resolution of MRI data. It should be included in the manuscript.
; Thanks for the comment. We added the detailed information of the MRI as below:
We used 3.0T brain MRI scanner (Skyra, Siemens, Germany) with the following parameters: TR, 9,000-11,000 range ; TE 120-130 range; matrix size, 256 × 256; field of view, 230 × 230 mm; slice thickness, 5mm; inter-slice gap, 1mm.
There is no information about the hardware applied in the analysis. It should be included in the manuscript.
; Thanks for the comment. We added detailed hardware information in the “ML Task” subsection.
Model training was performed on TensorFlow and Keras platform using NVIDIA’s GeForce GTX 1080ti graphics processing units, dual Xeon central processing units, and 128 GB RAM server.
Conclusions
More informative conclusions should be included.
; Thanks for the comment. We added some sentence for the informative future perspectives as the reviewer’s suggestion.
Figures
Figure 2 – the resolution should be improved.
Figure 3 – the resolution should be improved.
; Thanks for the comment. We uploaded those figures with high resolution as a separate file.

Round 2
Reviewer 2 Report
I accept the revised version of manuscript.